# Association between platelet distribution width and prognosis in patients with heart failure

Yu Sato[1], Akiomi Yoshihisa[1,2]*, Koichiro Watanabe[1], Yu Hotsuki[1], Yusuke Kimishima[1], Tetsuro Yokokawa[1], Tomofumi Misaka[1,2], Takamasa Sato[1], Takashi Kaneshiro[1], Masayoshi Oikawa[1], Atsushi Kobayashi[1], Yasuchika Takeishi[1]

1 Department of Cardiovascular Medicine, Fukushima Medical University, Fukushima, Japan, 2 Department of Advanced Cardiac Therapeutics, Fukushima Medical University, Fukushima, Japan

* yoshihis@fmu.ac.jp

**Data Availability Statement:** All relevant data are within the manuscript and its Supporting Information files.

## Abstract

### Background

The prognostic impact of platelet distribution width (PDW), which is a specific marker of platelet activation, has been unclear in patients with heart failure (HF).

### Methods and results

We conducted a prospective observational study enrolling 1,746 hospitalized patients with HF. Patients were divided into tertiles based on levels of PDW: 1st (PDW < 15.9 fL, n = 586), 2nd (PDW 15.9–16.8 fL, n = 617), and 3rd (PDW $\geq$ 16.9, n = 543) tertiles. We compared baseline patients' characteristics and post-discharge prognosis: all-cause death; cardiac death; and cardiac events. The 3rd tertile showed the highest age and levels of B-type natriuretic peptide compared to other tertiles (1st, 2nd, and 3rd tertiles; age, 69.0, 68.0, and 70.0 years old, P = 0.038; B-type natriuretic peptide, 235.2, 171.9, and 241.0 pg/mL, P < 0.001). Left ventricular ejection fraction was equivalent among the tertiles. In the Kaplan-Meier analysis, rates of all endpoints were the highest in the 3rd tertile (log-rank P < 0.001, respectively). The Cox proportional hazard analysis revealed that the 3rd tertile was associated with adverse prognosis (all-cause death, hazard ratio [HR] 1.716, P < 0.001; cardiac death, HR 1.919, P < 0.001; cardiac event, HR 1.401, P = 0.002).

### Conclusions

High PDW is a novel predictor of adverse prognosis in patients with HF.

## Introduction

Accurate risk stratification and estimation of prognosis are critically important in patients with heart failure (HF) [1, 2]. Current major guidelines recommend laboratory evaluation, including complete blood count [1, 2]. However, the usefulness of complete blood count other

**Funding:** This study was supported in part by a grant-in-aid for Scientific Research (No. 20K07828) from the Japan Society for the Promotion of Science. There was no additional external funding received for this study.

**Competing interests:** The authors have declared that no competing interests exist.

than hemoglobin or hematocrit in predicting prognosis has yet to be fully evaluated [1–3]. Red blood cell distribution width has been attracting great interest in recent years as a novel prognostic marker in patients with cardiovascular disease [4, 5]. Felker et al. firstly reported that higher red blood cell distribution width was independently and strongly associated with all-cause death in patients with HF [4]. Subsequently, Tonelli et al. analyzed data from a large cohort of patients with myocardial infarction, and found that red blood cell distribution width was also useful in predicting all-cause death [5]. Complete blood count routinely reports not only red blood cell distribution width, but also platelet distribution width (PDW). PDW is defined as the distribution width (femtoliter, fL) at 20% of the total height of the platelet size distribution curve, measures the variability in platelet size, and is a marker of platelet activation [6–8]. Increased levels of PDW are presumed to be associated with atherosclerosis, coronary artery disease (CAD), cerebrovascular disease, and systemic inflammatory disease [9–11]. These diseases play key roles in the pathophysiology of HF [1, 2, 12–14]. Moreover, levels of PDW predicts the occurrence of cardiac death, infarction recurrence, and another revascularization in patients with acute myocardial infarction [7, 15]. In addition to routine clinical laboratory tests, new biomarkers reflecting various pathophysiological aspects can be useful adjuncts for the diagnosis, prognosis prediction, and treatment of HF [1, 2]. Thus, the aim of the present study was to evaluate PDW as a prognostic marker in patients with HF.

## Methods

This was a prospective and observational study. The patient flowchart is shown in Fig 1. The inclusion criteria included patients who 1) were both hospitalized for acute decompensated HF at, and discharged from, Fukushima Medical University Hospital between October 2009 and September 2019; and (2) were measured for PDW. HF was diagnosed in accordance with the current guidelines [1, 2, 16]. A total of 2,450 patients met the criteria. Among them, patients who were receiving maintenance dialysis and/or those with a history of cancer, including blood cancer, were excluded (n = 704). The definition of cancer was in accordance with our previous study [17]. Finally, a total of 1,746 patients were enrolled. We divided these patients into tertiles according to PDW level: patients with PDW in the lowest tertile (1st tertile, PDW < 15.9 fL, n = 586, 33.6%); those with PDW in the middle tertile (2nd tertile, PDW 15.9–16.8 fL, n = 617, 35.3%); and those with PDW in the highest tertile (3rd tertile, PDW ≥ 16.9 fL, n = 543, 31.1%). We compared baseline characteristics and post-discharge prognosis. The baseline characteristics comprised demographic data at discharge,

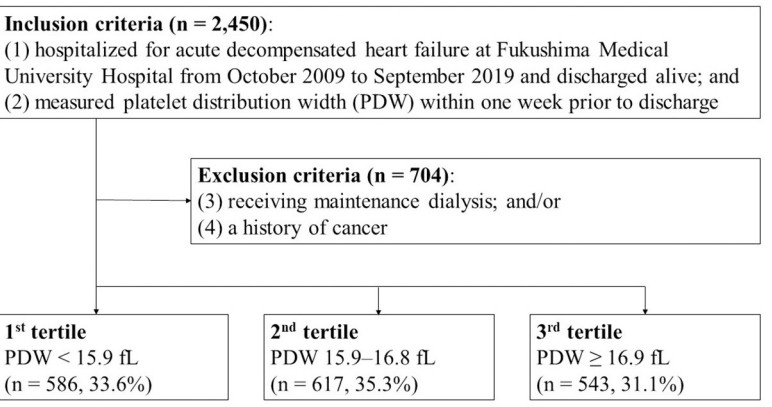

**Fig 1. Patient flowchart.** PDW, platelet distribution width.

comorbidities, medications at discharge, and laboratory and echocardiographic data obtained within 1 week prior to discharge in a stable condition. Comorbidities were defined in accordance with our previous studies [17, 18]. Anemia was defined as levels of hemoglobin of less than 13 g/dL in men and less than 12 g/dL in women [19]. CAD was defined by the following: medical history, myocardial scintigraphy, coronary CT angiography, and/or coronary angiography [18]. Chronic kidney disease was defined as an estimated glomerular filtration rate of less than 60 mL/min/1.73 $m^2$ using a three-variable Japanese equation [20, 21]. The primary endpoints were all-cause death, cardiac death, and cardiac event. A cardiac death was defined as a death from worsened HF, ventricular fibrillation, or acute coronary syndrome. A cardiac event was defined as a cardiac death or an unplanned hospitalization for HF. If two or more cardiac events occurred in one patient, the first event was used for analyses as a cardiac event. This study complied with the Declaration of Helsinki and the statement of STROBE (Strengthening the Reporting of Observational studies in Epidemiology) [22, 23]. The study protocol was approved by the ethical committee of Fukushima Medical University. All patients gave written informed consent to participate in this study.

PDW was analyzed from anticoagulated blood (ethylenediaminetetraacetic acid bulb) using an automated hematology analyzer, UniCel DxH 800 (BECKMAN COULTER, Brea, CA, USA) within 1 week prior to discharge in a stable condition. This procedure was performed by experienced laboratory technicians at Fukushima Medical University Hospital who were independent of this study.

Continuous variables were expressed as a median ($25^{th}$, $75^{th}$ percentile) because the Shapiro-Wilk test revealed that all the continuous variables analyzed in this study were non-normally distributed. Categorical variables were presented as counts (percent). Continuous variables were compared using the Kruskal-Wallis test followed by the Steel-Dwass post-hoc test. Categorical variables were compared using the chi-square test. The Kaplan-Meier analysis with log-rank test was used for presenting the primary endpoints. We assessed PDW as a predictor of the primary endpoints using the Cox proportional hazard analysis. Hazard ratio (HR) and 95% confidence interval (CI) were further adjusted for age, sex, and variables that were significantly different among the groups. The threshold for statistical significance was $P < 0.05$ for all the analyses. The Kruskal-Wallis test and the Steel-Dwass post-hoc test were performed using EZR version 1.40 (Saitama Medical Center, Jichi Medical University, Saitama, Japan) [24]. All other analyses were conducted using IBM SPSS Statistics version 26 (IBM, Armonk, NY, USA).

## Results

The distribution of patients according to PDW level is shown in Fig 2, and comparisons of the baseline patient characteristics are summarized in Table 1. The PDW levels in the $1^{st}$, $2^{nd}$, and $3^{rd}$ tertiles were 13.4 (12.2, 14.2), 16.5 (16.2, 16.7), and 17.3 (17.0, 17.7) fL, respectively ($P < 0.001$). There were U-shaped associations between PDW tertiles and several important factors such as age, hypertension, diabetes mellitus, coronary artery disease, and levels of B-type natriuretic peptide. On the other hand, the prevalence of chronic kidney disease was highest in the $3^{rd}$ tertile, followed by the $2^{nd}$, then $1^{st}$ tertiles (57.5%, 49.3%, and 49.0% respectively, $P = 0.005$). Left ventricular ejection fraction was equivalent among the groups ($P = 0.786$).

During the post-discharge follow-up period (median 1,352 days), there were 391 all-cause deaths including 212 cardiac deaths, and 480 cardiac events including 399 unplanned hospitalizations for HF and 81 cardiac deaths. The Kaplan-Meier analysis demonstrated that the rates of all the primary endpoints were highest in the $3^{rd}$ tertile (Fig 3, log-rank $P < 0.001$, respectively). Table 2 summarizes the results of the Cox proportional hazard analysis. In the

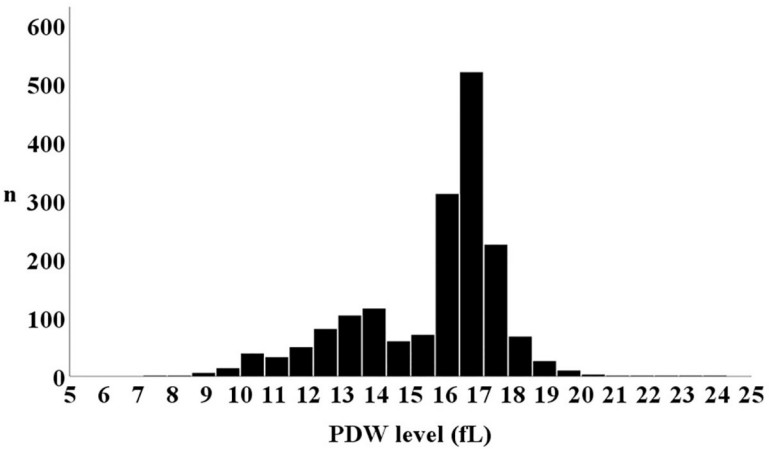

**Fig 2. Distribution of PDW.** PDW, platelet distribution width.

**Table 1. Baseline patient characteristics (n = 1,746).**

| | 1st tertile (n = 586) | 2nd tertile (n = 617) | 3rd tertile (n = 543) | P value |
|---|---|---|---|---|
| **PDW, fL** | 13.4 (12.2, 14.2) | 16.5 (16.2, 16.7) * | 17.3 (17.0, 17.7) *† | < 0.001 |
| **Demographic data** | | | | |
| Age, years old | 69.0 (58.0, 77.0) | 68.0 (56.0, 77.0) | 70.0 (59.0, 78.0) † | 0.038 |
| Male sex, n (%) | 353 (60.2) | 358 (58.0) | 330 (60.8) | 0.592 |
| Body mass index, kg/m$^2$ | 23.2 (21.0, 26.2) | 22.9 (20.6, 25.5) | 22.8 (20.6, 25.9) | 0.273 |
| Smoking history, n, (%) | 301 (52.4) | 300 (49.5) | 294 (54.4) | 0.241 |
| **Comorbidity** | | | | |
| Hypertension, n (%) | 451 (77.0) | 372 (60.3) | 346 (63.7) | < 0.001 |
| Diabetes mellitus, n (%) | 236 (40.3) | 188 (30.5) | 232 (42.7) | < 0.001 |
| Dyslipidemia, n (%) | 451 (77.0) | 390 (63.2) | 378 (69.6) | < 0.001 |
| Hyperuricemia, n (%) | 341 (58.2) | 335 (54.3) | 339 (62.4) | 0.020 |
| Anemia, n (%) | 283 (48.3) | 254 (41.2) | 285 (52.5) | < 0.001 |
| Atrial fibrillation, n (%) | 206 (35.2) | 236 (38.2) | 218 (40.1) | 0.215 |
| CAD, n (%) | 225 (38.4) | 159 (25.8) | 166 (30.6) | < 0.001 |
| CVA, n (%) | 105 (17.9) | 93 (15.1) | 93 (17.1) | 0.392 |
| PAD, n (%) | 65 (18.3) | 56 (15.1) | 51 (16.3) | 0.512 |
| CKD, n (%) | 287 (49.0) | 304 (49.3) | 312 (57.5) | 0.005 |
| COPD, n (%) | 129 (28.2) | 153 (27.8) | 140 (30.4) | 0.643 |
| **Medication** | | | | |
| RAS inhibitors, n (%) | 450 (76.8) | 419 (67.9) | 376 (69.2) | 0.001 |
| Beta blockers, n (%) | 435 (74.2) | 439 (71.2) | 379 (69.8) | 0.233 |
| Loop diuretics, n (%) | 393 (67.1) | 398 (64.5) | 380 (70.0) | 0.141 |
| Anticoagulants, n (%) | 321 (54.8) | 376 (60.9) | 311 (57.3) | 0.093 |
| Antiplatelet agents, n (%) | 329 (56.1) | 301 (48.8) | 274 (50.5) | 0.029 |
| **BNP (pg/mL)** | 235.2 (64.8, 602.0) | 171.9 (66.9, 458.9) * | 241.0 (92.0, 568.3) † | < 0.001 |
| **LVEF (%)** | 53.0 (39.0, 63.3) | 55.5 (39.0, 64.2) | 54.7 (39.1, 64.4) | 0.786 |

PDW, platelet distribution width; CAD, coronary artery disease; CVA, cerebrovascular accident; PAD, peripheral artery disease; CKD, chronic kidney disease; COPD, chronic obstructive pulmonary disease; RAS, renin-angiotensin system; BNP, B-type natriuretic peptide; LVEF, left ventricular ejection fraction.

*P < 0.05 vs. 1st tertile and

†P < 0.05 vs. 2nd tertile.

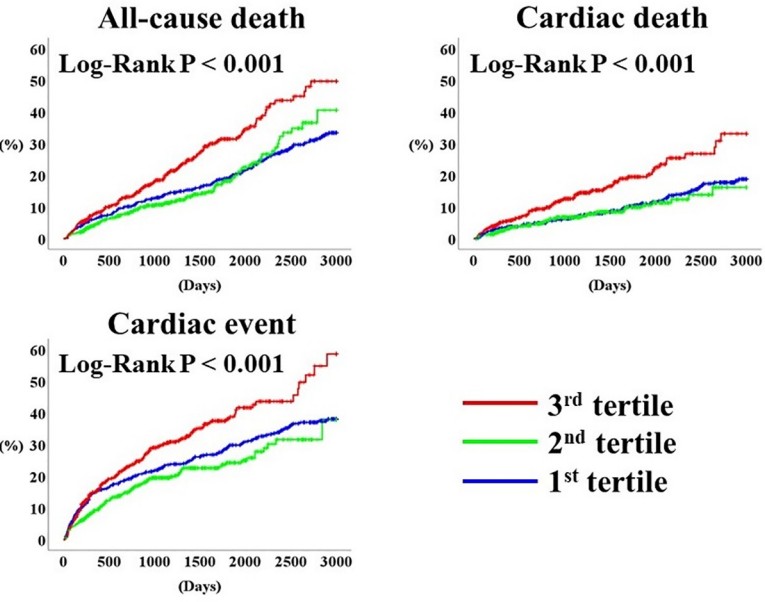

**Fig 3. Prognosis stratified by PDW.** PDW, platelet distribution width.

unadjusted analysis, the 3rd tertile (vs. 2nd and 1st tertiles) were associated with all primary endpoints. In the multivariate analysis, the 3rd tertile was independently associated with adverse prognosis compared to the 2nd and 1st tertiles.

Considering the results from multivariate Cox proportional hazard analysis, we performed the logistic regression analysis to determine the factors associated with high (3rd tertile) PDW (S1 Table). Age, sex and factors which differed significantly in Table 1 were evaluated. From

**Table 2. Cox proportional hazard analysis: The prognostic impact of PDW.**

| | Unadjusted | | Adjusted * | |
|---|---|---|---|---|
| | HR (95% CI) | P value | HR (95% CI) | P value |
| **All-cause death (event n = 391/ 1,746)** | | | | |
| 2nd tertile (vs. 1st tertile) | 1.012 (0.779–1.314) | 0.930 | – | – |
| 3rd tertile (vs. 1st tertile) | 1.716 (1.356–2.172) | < 0.001 | 1.301 (1.003–1.688) | 0.047 |
| 3rd tertile (vs. 2nd tertile) | 1.696 (1.310–2.196) | < 0.001 | 1.260 (0.959–1.656) | 0.097 |
| **Cardiac death (event n = 212/ 1,746)** | | | | |
| 2nd tertile (vs. 1st tertile) | 0.924 (0.640–1.334) | 0.672 | – | – |
| 3rd tertile (vs. 1st tertile) | 1.919 (1.400–2.630) | < 0.001 | 1.421 (1.006–2.008) | 0.046 |
| 3rd tertile (vs. 2nd tertile) | 2.077 (1.459–2.957) | < 0.001 | 1.537 (1.059–2.232) | 0.024 |
| **Cardiac event (event n = 480/ 1,746)** | | | | |
| 2nd tertile (vs. 1st tertile) | 0.831 (0.659–1.048) | 0.117 | – | – |
| 3rd tertile (vs. 1st tertile) | 1.401 (1.133–1.732) | 0.002 | 1.168 (0.926–1.472) | 0.189 |
| 3rd tertile (vs. 2nd tertile) | 1.686 (1.339–2.123) | < 0.001 | 1.296 (1.021–1.645) | 0.033 |

HR, hazard ratio; CI, confidence interval; PDW, platelet distribution width.

* Adjusted for age, sex, presence or absence of hypertension, diabetes mellitus, dyslipidemia, hyperuricemia, anemia, coronary artery disease, and chronic kidney disease, use of renin-angiotensin system inhibitors and antiplatelet agents, and log-transformed B-type natriuretic peptide.

the multivariate model, diabetes mellitus and anemia were independently associated with high PDW with adjusted odds ratios of 1.352 and 1.275, respectively (S1 Table).

## Discussion

To the best of our knowledge, the current study is the first to report that high PDW is an independent predictor of adverse prognosis in patients with HF. PDW tertiles showed U- or J-shaped associations with HF severity and prognosis, which suggests that there may be a threshold in PDW level for identifying patients who are at high risk of HF. These findings are pivotal for clinical practice because complete blood count is performed in almost all patients with HF, and PDW is reported routinely and automatically.

PDW and mean platelet volume are positively correlated, and both are simple parameters of platelet activation [25, 26]. PDW is a more specific marker of platelet activation than mean platelet volume, since it does not increase during simple platelet swelling [7, 27]. Growth factors and cytokines in the atherosclerotic processes may elevate levels of PDW through interference with platelet production in the bone marrow [28, 29]. Larger platelets are metabolically and enzymatically more active than smaller platelets [30]. Activated platelets stimulate thrombus formation, promoting atherothrombotic disease in response to rupture of an atherosclerotic plaque or endothelial cell erosion [31]. However, the clinical association between levels of PDW and CAD has been controversial. Higher PDW was previously reported to be independently associated with a high SYNTAX score, chronic total occlusion, inadequate coronary collateral, and in-stent restenosis in patients with CAD [8, 26, 29, 32]. On the other hand, De Luca et al. reported that PDW was not associated with CAD prevalence [33]. In terms of prognosis, higher levels of PDW have been reported to independently predict cardiac death and adverse cardiovascular events in patients with CAD [7, 34]. One plausible explanation for the discrepancy is the presence of the threshold shown in our results, which were consistent with those of a previous report [35].

According to He et al., who analyzed big data on retired employees, there were non-linear associations between levels of PDW and prognosis in terms of cardiovascular disease, CAD, and stroke [36]. In their report, participants with the lowest quintile had better prognosis compared to those with PDW in the 20–80th percentiles [36]. Considering the non-linear associations, PDW should be handled as a categorical variable divided by certain thresholds for prognosis prediction. In addition, there were significant differences regarding participant characteristics between our study and the above-mentioned studies, which suggests that the prognostic impact of PDW may differ according to the target population (e.g. patients with HF, those with CAD, or general population).

Although there have been numerous biomarkers reported in patients with HF, accurate risk stratification remains challenging [1, 2]. Natriuretic peptide (NP) is widely used to support diagnosis, prognostication, and management of patients with HF, but NPs come with limitations, including vulnerability to the presence of obesity, atrial fibrillation, and renal dysfunction, for example [37]. Aside from NPs, novel biomarkers may improve the understanding of the complex disease process of HF, and possibly personalize care for those affected through better individual phenotyping or understanding pathophysiologic pathways (e.g. myocardial stretch/stress, cardiac extracellular matrix remodeling, cardiomyocyte injury/death, oxidative stress, inflammation, neurohumoral activation, and renal dysfunction) [37]. In these regards, it is reasonable to expect the application of a multibiomarker approach for the improvement in management of HF and the personalization of care. From the results of our study, the presence of diabetes mellitus and anemia, but not B-type natriuretic peptide, was associated with elevated levels of PDW. This suggested that the underlying mechanisms of high PDW were

different from left ventricular stretch or diastolic wall stress [38, 39]. Levels of PDW show a positive correlation with hemoglobin A1c [40]. Patients with diabetes mellitus, particularly those with microvascular complications, show increased levels of PDW [41]. Previous studies suggested that the prothrombotic state due to insulin resistance and insulin deficiency may have contributed to increasing platelet reactivity [41, 42]. Patients with diabetes mellitus demonstrate increased platelet activation and increased release of prothrombotic and proinflammatory agents due to systematic inflammation, oxidative stress, impaired calcium metabolism, decreased bioavailability of nitric oxide, increased phosphorylation and glycosylation of cellular proteins [43]. More active platelets have more prothrombic contents, such as thromboxane A2, thromboxane B2, platelet factor 4, serotonin, and platelet-derived growth factor [44]. Thus, HF patients with increased PDW are suspected to have these underlying pathophysiologies. On the other hand, anemia is associated with myocardial remodeling, inflammation, and volume overload in patients with HF [45]. However, the link between anemia and PDW has been unclear. Levels of PDW depend on whether platelets are hypoproductive such as aplastic anemia or hyperdestructive [46]. The clinical implications of PDW in anemia should be investigated by further studies because the leading cause of anemia was not recorded in this study. Thus, PDW is useful for multifaceted assessment of the pathophysiology of HF.

An unexpected finding of our results was that patients in the 1$^{st}$ tertile showed several worse characteristics compared to those in the 2$^{nd}$ tertile. Levels of B-type natriuretic peptide were obviously U-shaped. We were not able to explain the differences between patients in the 1$^{st}$ and 2$^{nd}$ tertiles due to the study protocol. However, after adjustment for important prognostic factors, the 3$^{rd}$ PDW tertile showed the worst outcome among the tertiles. Although there have been various studies on the association between PDW and CAD, evidence for PDW on HF has been little. Further studies are needed to clarify the clinical significance of PDW in patients with HF.

### Study limitations

Due to the relatively small number of patients of this present study, the accurate PDW cut-off level should be investigated in further studies with larger populations. The underlying mechanisms influencing the levels of PDW need more investigation. The analyzed PDW levels were measured within 1 week prior to discharge; changes in PDW through the clinical course were not considered. There were some patients receiving anticoagulants for reasons other than atrial fibrillation (e.g. post-operatic state). However, the leading cause of anticoagulant therapy was not taken into consideration in this study.

### Conclusions

High PDW is an independent predictor of adverse prognosis in patients with HF.

### Supporting information

**S1 Table. Logistic regression analysis for the 3$^{rd}$ tertile (n = 543/1,746).** OR, odds ratio; CI confidence interval; CAD, coronary artery disease; CKD, chronic kidney disease; RAS, renin-angiotensin system; Log-BNP, log-transformed B-type natriuretic peptide. *Adjusted for age, sex, and factors which had P values of < 0.05 in the unadjusted model.
(DOCX)

**S1 Dataset.**
(SAV)

## Acknowledgments

The authors thank Ms. Kumiko Watanabe, Ms. Hitomi Kobayashi, Ms. Yumi Yoshihisa, and Ms. Tomiko Miura for their technical assistance.

## Author Contributions

**Conceptualization:** Yu Sato, Akiomi Yoshihisa, Koichiro Watanabe, Yu Hotsuki, Yusuke Kimishima, Tetsuro Yokokawa, Tomofumi Misaka, Takamasa Sato, Takashi Kaneshiro, Masayoshi Oikawa, Atsushi Kobayashi, Yasuchika Takeishi.

**Data curation:** Yu Sato, Akiomi Yoshihisa.

**Formal analysis:** Yu Sato, Akiomi Yoshihisa, Koichiro Watanabe, Yu Hotsuki, Yusuke Kimishima, Tetsuro Yokokawa, Tomofumi Misaka, Takamasa Sato, Takashi Kaneshiro, Masayoshi Oikawa, Atsushi Kobayashi, Yasuchika Takeishi.

**Funding acquisition:** Akiomi Yoshihisa.

**Investigation:** Yu Sato, Akiomi Yoshihisa, Koichiro Watanabe, Yu Hotsuki, Yusuke Kimishima, Tetsuro Yokokawa, Tomofumi Misaka, Takamasa Sato, Takashi Kaneshiro, Masayoshi Oikawa, Atsushi Kobayashi, Yasuchika Takeishi.

**Methodology:** Yu Sato, Akiomi Yoshihisa, Koichiro Watanabe, Yu Hotsuki, Yusuke Kimishima, Tetsuro Yokokawa, Tomofumi Misaka, Takamasa Sato, Takashi Kaneshiro, Masayoshi Oikawa, Atsushi Kobayashi, Yasuchika Takeishi.

**Project administration:** Akiomi Yoshihisa, Yasuchika Takeishi.

**Resources:** Akiomi Yoshihisa.

**Supervision:** Akiomi Yoshihisa, Yasuchika Takeishi.

**Visualization:** Yu Sato, Akiomi Yoshihisa.

**Writing – original draft:** Yu Sato, Akiomi Yoshihisa, Yasuchika Takeishi.

**Writing – review & editing:** Koichiro Watanabe, Yu Hotsuki, Yusuke Kimishima, Tetsuro Yokokawa, Tomofumi Misaka, Takamasa Sato, Takashi Kaneshiro, Masayoshi Oikawa, Atsushi Kobayashi.

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
