## [Decision Letter · Decision Letter 0]

22 Sep 2020

PONE-D-20-22248

Association between Platelet Distribution Width and Prognosis in Patients with Heart Failure

PLOS ONE

Dear Dr. Yoshihisa,

Thank you for submitting your manuscript to PLOS ONE. After careful consideration, we feel that it has merit but does not fully meet PLOS ONE’s publication criteria as it currently stands. Therefore, we invite you to submit a revised version of the manuscript that addresses the points raised during the review process.

Please note that the manuscript has important limitations, which make it not acceptable in its present form. Specifically, the question arises, as also addressed by the reviewer, why additional prognostic markers are needed in heart failure. in fact, I do not see any reasons unless a novel marker is extremely outstanding in this regard, which is not the case. There are already so many and in clinical practice, the impact even of the most established ones (e.g. NTpro-BNP / BNP) is unclear. Therefore, the findings of a new prognostic marker is scientifically not interesting, unless the study provides novel pathophysiological insight. Unfortunately, this is hardly addressed in the manuscript, which is a very important limitation of the manuscript. Additional analyses addressing potential pathophysiological mechanisms may help in this regard. The authors only show differences in baseline characteristics between tertiles of PWD, but do not address factors potentially influencing PWD. This could help to estimate the meaning and as a consequence, also the potential value of PWD. This needs to be properly addressed. In addition, there are minor issues to solve. Thus, the authors should avoid to repeat results in the text that are already given in tables or figures. The authors need to make clear, why they did not include more co-morbidities and also not including them in the multivariable model. Moreover, the authors do not consider potential dependencies between variables in multivariable models. The language of the manuscript needs to be improved as well.

Take together, a substantial change of the manuscript and the analysis is required before the manuscript may be considered for publication. I would like to leave it to the authors if they are willing to perform such substantial changes. If not, the manuscript will be rejected, however.

We look forward to receiving your revised manuscript.

Kind regards,

Hans-Peter Brunner-La Rocca, M.D.

Academic Editor

PLOS ONE

Journal Requirements:

Reviewers' comments:

Reviewer's Responses to Questions

**Comments to the Author**

1. Is the manuscript technically sound, and do the data support the conclusions?

Reviewer #1: Yes

2. Has the statistical analysis been performed appropriately and rigorously? 

Reviewer #1: Yes

3. Have the authors made all data underlying the findings in their manuscript fully available?

Reviewer #1: Yes

4. Is the manuscript presented in an intelligible fashion and written in standard English?

Reviewer #1: No

5. Review Comments to the Author

Reviewer #1: Line 29: Age was oldest. should be rewritten. This should also be modified further along in the manuscript

Introduction: Although the authors mention the CBC, they do not mention the value of hematocrit which has been studied as a prognostic marker in HF.

Table 1: although the prevalence of AFib in this patient population was close to 40%, anticoagulation was used in 54-60% of patients. Could the authors explain the discrepancy?

Table 1: LVEF was around 50%. Does this actually reflect the population normally admitted to hospital with acute decompensated HF? Previous studies of all hospitalizations for HF usually enroll patients with a mean EF slightly lower.

Line 177: The authors does raise an important question regarding the discrepancy of some of their findings. Additional work is needed to better understand this finding.

Overall, although these findings are interesting, I am not convinced that we need an additional prognostic marker for HF when strong biomarkers such as BNP and NT proBNP are most often available.

6. PLOS authors have the option to publish the peer review history of their article (what does this mean?). If published, this will include your full peer review and any attached files.

Reviewer #1: No

---

## [Editor Report · Decision Letter 1]

27 Nov 2020

PONE-D-20-22248R1

Association between platelet distribution width and prognosis in patients with heart failure

PLOS ONE

Dear Dr. Yoshihisa,

Thank you for submitting your manuscript to PLOS ONE. After careful consideration, we feel that it has merit but does not fully meet PLOS ONE’s publication criteria as it currently stands. Therefore, we invite you to submit a revised version of the manuscript that addresses the points raised during the review process.

Please not the comment that I have written below.

We look forward to receiving your revised manuscript.

Kind regards,

Hans-Peter Brunner-La Rocca, M.D.

Academic Editor

PLOS ONE

Additional Editor Comments (if provided):

I would like to thank the authors for addressing most of the raised points. However, there is still a remaining issue that may be improved. Although I know that the pathophysiology of PDW is not yet investigated in much detail, the authors should address the pathophysiological part of finding a new biomarker in general and specifically for PDW more in depth. As mentioned in the previous comments, simply finding a new marker is not of much value, even if such a marker is easy to determine. It is not only (NT-pro)BNP which is a good prognostic marker, but there are so many other markers (including biomarkers) found that are related to outcome that a new one must be exceptionally good in predicting outcome (which is not the case here) or help the reader to understand a (new) pathophysiological process.

---

## [Editor Report · Decision Letter 2]

14 Dec 2020

Association between platelet distribution width and prognosis in patients with heart failure

PONE-D-20-22248R2

Dear Dr. Yoshihisa,

We’re pleased to inform you that your manuscript has been judged scientifically suitable for publication and will be formally accepted for publication once it meets all outstanding technical requirements.

Kind regards,

Hans-Peter Brunner-La Rocca, M.D.

Academic Editor

PLOS ONE
---

## [Editor Report · Acceptance letter]

16 Dec 2020

PONE-D-20-22248R2 

Association between platelet distribution width and prognosis in patients with heart failure 

Dear Dr. Yoshihisa:

I'm pleased to inform you that your manuscript has been deemed suitable for publication in PLOS ONE. Congratulations! Your manuscript is now with our production department. 

Kind regards, 

on behalf of

Dr. Hans-Peter Brunner-La Rocca 

Academic Editor

PLOS ONE